# Association Mapping of Candidate Genes Associated with Iron and Zinc Content in Rice (*Oryza sativa* L.) Grains

**DOI:** 10.3390/genes14091815

**Published:** 2023-09-18

**Authors:** Chance Bahati Bukomarhe, Paul Kitenge Kimwemwe, Stephen Mwangi Githiri, Edward George Mamati, Wilson Kimani, Collins Mutai, Fredrick Nganga, Paul-Martin Dontsop Nguezet, Jacob Mignouna, René Mushizi Civava, Mamadou Fofana

**Affiliations:** 1Department of Horticulture and Food Security, Jomo Kenyatta University of Agriculture and Technology (JKUAT), Nairobi P.O. Box 62000-00200, Kenya; kimwemwe.paul@students.jkuat.ac.ke (P.K.K.); githirim@agr.jkuat.ac.ke (S.M.G.); egmamati@agr.jkuat.ac.ke (E.G.M.); 2Olusegun O. Research Campus, International Institute of Tropical Agriculture (IITA), Bukavu P.O. Box 1222, Democratic Republic of the Congo; j.mignouna@cgiar.org (J.M.); ma.fofana@cgiar.org (M.F.); 3Institut National Pour l’Etude et la Recherche Agronomiques (INERA), Kinshasa P.O. Box 2037, Democratic Republic of the Congo; renecivava@uea.ac.cd; 4Faculty of Agriculture and Environmental Sciences, Université de Kalemie (UNIKAL), Kalemie P.O. Box 570, Democratic Republic of the Congo; 5International Livestock Research Institute (ILRI), Nairobi P.O. Box 30709-00100, Kenya; c.mutai@cgiar.org (C.M.); f.nganga@cgiar.org (F.N.); 6International Institute of Tropical Agriculture (IITA), Kalemie P.O. Box 570, Democratic Republic of the Congo; p.dontsop@cgiar.org; 7Faculty of Agriculture and Environmental Sciences, Université Evangélique en Afrique (UEA), Bukavu P.O. Box 3323, Democratic Republic of the Congo

**Keywords:** rice, Fe and Zn content, genetics, SNP markers, GWAS, gene identification

## Abstract

Micronutrient deficiencies, particularly of iron (Fe) and zinc (Zn), in the diet contribute to health issues and hidden hunger. Enhancing the Fe and Zn content in globally staple food crops like rice is necessary to address food malnutrition. A Genome-Wide Association Study (GWAS) was conducted using 85 diverse rice accessions from the Democratic Republic of Congo (DRC) to identify genomic regions associated with grain Fe and Zn content. The Fe content ranged from 0.95 to 8.68 mg/100 g on a dry weight basis (dwb) while Zn content ranged from 0.87 to 3.8 mg/100 g (dwb). Using MLM and FarmCPU models, we found 10 significant SNPs out of which one SNP on chromosome 11 was associated with the variation in Fe content and one SNP on chromosome 4 was associated with the Zn content, and both were commonly detected by the two models. Candidate genes belonging to transcription regulator activities, including the bZIP family genes and MYB family genes, as well as transporter activities involved in Fe and Zn homeostasis were identified in the vicinity of the SNP markers and selected. The identified SNP markers hold promise for marker-assisted selection in rice breeding programs aimed at enhancing Fe and Zn content in rice. This study provides valuable insights into the genetic factors controlling Fe and Zn uptake and their transport and accumulation in rice, offering opportunities for developing biofortified rice varieties to combat malnutrition among rice consumers.

## 1. Introduction

The global population is projected to reach approximately 9.7 billion by 2050 [1], and a significant proportion of it faces a major problem of malnutrition. Malnutrition manifests itself in various forms and it impacts not only on an individual’s health and well-being but also imposes significant burdens on families, communities, and states [2]. It continues to be a significant public health concern in many developing countries [3] in the world, with the African continent being the most affected. In the Democratic Republic of Congo (DRC), micronutrient (iron and zinc) deficiencies contribute to a substantial number of deaths in women and children [4]. Iron (Fe) is vital for various biological functions in the human body, including the synthesis of oxygen transport proteins [5,6]. Its deficiency leads to anemia [7] while zinc (Zn) deficiency affects multiple systems in the human body, including epidermal, gastrointestinal, central nervous, immune, skeletal, and reproductive systems [8].

Rice (*Oryza sativa* L.) is the most important food crop, feeding more than half of the world’s population [9]. Other than satisfying the caloric requirements of millions of people daily [10], rice serves as the primary source of protein, thiamine, riboflavin, niacin, and essential micronutrients, such as Fe and Zn in the diet [9,11]. However, rice has been reported to have a relatively low content of Fe and Zn, particularly when paddies undergo processing [12].

Meeting the food and nutritional needs of a growing population presents a significant challenge, especially considering the imminent climatic variability and limited resources. Adequate consumption of fortified food with essential micronutrients, such as Fe and Zn, is crucial for individuals to meet their metabolic requirements to maintain optimal health and reduce hidden hunger [13]. This can be achieved by enhancing Fe and Zn content in the most consumed foods. 

Plants play a critical role as pathways for nutrient movement from soil to humans [14]. More efforts were deployed by the Consultative Group on International Agricultural Research (CGIAR) institutes worldwide, such as the International Food Policy Research Institute (IFPRI) and the International Centre for Tropical Agriculture (CIAT) under the HarvestPlus program in developing and introducing biofortified varieties of crops [15]. In this context, cultivars of beans, maize, sweet potato, and cassava have been introduced into the DRC to alleviate hidden hunger and reduce its consequences [16]. 

However, being the second most consumed cereal after maize [17], rice could constitute an alternative source of micronutrients in the DRC. Iron and Zinc content in grain rice are variable depending upon its variety and processing. The target levels for Fe and Zn content in polished rice grains, according to HarvestPlus, should be 1.3 and 2.4 mg/100 g, respectively [18].

Plants have specialized mechanisms for the uptake and accumulation of Fe and Zn from the soil. They secrete phytosiderophores (PSs) to chelate Fe(III) and form Fe(III)-PS complexes, which are then transported into the plant through specific plasma membrane transporters [19]. Rice also has mechanisms to directly uptake Fe(II) and chelated Zn, using the mugineic acid (MA) family. Zinc uptake occurs through specific plasma membrane transporters, and it can also enter the root directly as ionized Zn(II) [20].

The genetic control for metal transport has been studied in rice and *Arabidopsis,* which identified the founding members of metal transporters, specifically focusing on Fe and Zn. These include the zinc-regulated transporter/iron-regulated transporter (ZRT/IRT)-related protein (ZIP) family [21], the natural resistance-associated macrophage protein (NRAMP) family, the cation diffusion facilitator (CDF) family, the major facilitator superfamily (MFS), the P_1B_-type heavy metal ATPase (HMA) family, the vacuolar Fe transporter (VIT) family, and the cation exchange (CAX) family [22].

The identification of the molecular factors influencing mineral uptake and transport as well as understanding the genes that regulate mineral homeostasis and localization in rice, would greatly contribute to the development of focused breeding approaches for biofortification purposes. A Genome-Wide Association Study (GWAS) is a highly effective tool for elucidating the genetic basis of traits in plants [23]. It has been widely used in identifying quantitative trait loci (QTL) for grain quality traits, including the Fe and Zn content in rice [20,24,25], in wheat [26], in *Aegilops tauschii*, the wild progenitor of the bread wheat [27] and in beans [28].

This study aimed to identify genomic regions associated with the Fe and Zn content in rice grains, using SNP markers generated from a set of 85 diverse rice accessions, including local landraces. The results of this study present significant assets that can be utilized for breeding and improvement programs aimed at developing rice lines with high grain Fe and Zn content by integrating the identified markers into a Marker Assisted Selection (MAS) approach. This could enhance the nutritional value of rice and contribute to improved health outcomes in the DRC.

## 2. Materials and Methods

### 2.1. Plant Materials and Field Experiment Details

A set of 85 rice accessions (Appendix A) maintained at the rice breeding program of the Institut National pour l’Etude et la Recherche Agronomiques (INERA), in Eastern DRC, were used in this study. The accessions were grown in Kalemie (S05°49.770′ and E029°17.442′, altitude: 778 m) and in the Ruzizi plain (S03°04.360′ and E029°08.243′, altitude: 910 m) under irrigated conditions. The experiment was laid out in a 17 × 5 lattice design with three replications. The soils at kalemie are clay loam soils as well as in the Ruzizi plain. Two fractions of urea (46% N) at a rate of 60 kg ha^−1^ were utilized for fertilization purposes. The Fe and Zn concentration in the soils were 178.26 mg/kg and 17.62 mg/kg; and 118.2 mg/kg and 8 mg/kg, at both the Kalemie and Ruzizi plain, respectively. Samples from the two sites were pooled for micronutrient analysis.

### 2.2. Fe and Zn Content Evaluation

Fifteen grams (g) of paddy rice were sampled from each accession, dehusked, and milled for Fe and Zn concentration analysis at the Mycotoxin and nutrition platform at the International Livestock Research Institute (ILRI), Nairobi, Kenya. The AOAC official method 985.01 for metals and other elements in plants and pet foods, as described by Hou et al. [29], was used. In brief, 500 mg of the milled rice grain samples were weighed in duplicates into 50 mL microwave digestion tubes, then 8.0 mL of concentrated nitric acid and 2.0 mL of 30% hydrogen peroxide were added. The digestion was performed using an Anton Paar Multiwave GO plus microwave digestor (Graz, Austria). The samples were heated at 100 °C for 10 min, the temperature was then increased to 180 °C at a rate of 10 °C/min followed by a 10 min hold. The samples were quantitatively transferred to 25.0 mL flasks and topped to the mark with 2% HNO_3_. The extracts were analyzed using the Perkin Elmer Avio 550 Max ICP-OES instrument. For the quantification of Fe and Zn in the extract, ICP-OES mix standard CatNo.43843 (Sigma-Aldrich, Buchs, Switzerland) was used. The serial dilution of the standard was performed using 2% HNO_3_ to obtain the calibration standards of 80, 320, 800, and 1600 µg/L, and the external standard calibration method was then applied. The calibration was performed using Perkin Elmer syngistix^TM^ software version 5.1. The obtained data were used to calculate the content for each element in mg/100 g using Microsoft^®^ Excel^®^ (2018) by applying the formulae below:$$X(mg/100g)=\frac{\left(C-B\right)*V*100}{W*1000}$$
where X is the individual elemental composition, mg/100 g; C is the concentration of the individual elements, µg/L after external calibration; B is the concentration of the reagent blanks, µg/l used in the extraction; V is the volume digest topped up to 25 mL; 100 is the conversion factor to mg/100 g dilution factor after extraction with 1% HNO_3_; W is the weight of the sample used; 1000 is the conversion factor from µg/l to mg/l. The results were corrected for moisture content and reported on the dry weight basis (dwb).

For Quality Control (QC) purposes after every batch of 30 samples (Appendix A), a QC sample T18106QC (infant formula) obtained from Fera Science Ltd. Sand Hutton, York. YO41 1LZ. UK was subjected to the whole pipeline of analysis in duplicate.

### 2.3. Genotypic Data Acquisition and Analysis

Genotypic data were acquired using DArTseq technology, as previously described by Kimwemwe et al. [30]. Briefly, this procedure utilized the DArTseq complexity reduction approach for library construction, which involved digesting genomic DNA from rice accessions with two restriction enzymes (*PstI* and *MseI*), ligating barcoded adapters, and amplifying the adapter-ligated fragments using polymerase chain reaction (PCR) to generate a library of DNA fragments for sequencing [31].

The libraries were then sequenced with HiSeq2500 (Illumina, San Diego, CA, USA) and the resulting sequences were scored for DArTseq markers using an in-house marker scoring pipeline, DArTsoft14. The sequenced reads were then aligned to the rice reference genome version 7.0, from the Rice Genome Annotation Project (RGAP) database [32] to detect Single Nucleotide Polymorphism (SNP) markers and their corresponding chromosome and physical positions.

### 2.4. Evaluation of Linkage Disequilibrium Decay

The genome-wide linkage disequilibrium pattern for our diverse panel, based on the retained SNP markers after filtering, were estimated using Trait Analysis by aSSociation Evolution, and Linkage (TASSEL) software version 5.2.88 [33]. Pairwise association among all the SNP markers were calculated to obtain the correlation coefficient (r^2^). The averages of all the r^2^ across each of the 12 rice chromosomes were plotted against the physical distance of the SNPs using R software [34] to estimate the Linkage Disequilibrium (LD) decay simulation curve.

### 2.5. Fe and Zn Content Data Analysis

The distribution of the evaluated rice accessions based on Fe and Zn content was plotted using rcompanion package of R software [35]. The descriptive statistics including the median, mean, range, and the standard deviation were computed using the moments package of R software [36]. To understand the central tendency in the dataset used in this study, the mode was calculated using the dplyr package in R software [37]. The average values from the two replicates for each micronutrient were used for the Genome-Wide Association Analysis.

### 2.6. Genome-Wide Association Analysis

Genome-Wide Association Studies (GWAS) for each micronutrient was performed with rMVP package of R software [38] using two linear models; Mixed Linear Model (MLM) and fixed and random model circulating probability unification (FarmCPU). According to VanRaden [39], the kinship matrix and the principal components (top five) internally generated by the rMVP package were used as covariates within the rMVP package. The association between Fe and Zn content and genotype data were visualized using Manhattan plots. To obtain the estimated number of independent tests, the total length of the chromosomes (43,225,920 bp) was divided by the average LD decay distance (401,947 bp) observed in the association panel utilized for this study, resulting in 107.54 independent tests. To obtain a type I error probability of 0.05, a significance threshold was calculated by dividing 0.05 by the estimated number of independent tests (107.54) to obtain a threshold of 4.649 × 10^−4^.

### 2.7. Candidate Genes Identification and Gene Expression Analysis

After the GWAS, candidate genes for Fe and Zn content in rice grains were identified and annotated from the *Oryza sativa* reference genome version 7.0, available in the MSU-Rice Genome Annotation Project (RGAP) database [32]. All the genes within the LD decay window, upstream and downstream, of the significant SNPs were treated as potential candidate genes associated with Fe and Zn content. Based on functional annotations of the identified genes, the genes belonging to transporter activity and transcription regulator activity involved in Fe and Zn homeostasis were identified and selected. Publicly available RNA-seq data for rice found in Rice Expression Database (RED) [40] were mined for potential expression changes across all the selected candidate genes. 

The expression levels of the candidate genes were investigated by performing an in silico analysis of gene expression in different rice seed-related tissues, including aleurone tissues, anthers, panicles, pistils, and seeds. The expression levels in seed tissue were measured in units of Fragments per Kilobase Million (FPKM) and categorized based on the scale by Davidson et al. [41]: FPKM values ≤ 1 were classified as “not expressed,” FPKM ≤ 4 = “low-expressed,” FPKM values between 4 and 24 = “intermediate-expressed,” and FPKM values ≥ 24 designated as “high-expressed.”

## 3. Results

### 3.1. Fe and Zn Content Variations

The Fe content in rice grains showed a continuous variation but did not follow the normal distribution while the Zn content showed a normal distribution (Figure 1). The Fe content ranged between 0.95 and 8.68 mg/100 g (dwb) with a mean of 2.58 ± 1.31 mg/100 g (dwb). The Zn concentration ranged between 0.87 and 3.8 mg/100 g (dwb), with an average of 2.18 ± 0.35 mg/100 g (dwb) (Appendix A).

### 3.2. Genetic Markers Distribution and Linkage Disequilibrium Analysis

After filtering, 8379 polymorphic SNPs were retained and used for the GWAS and LD decay analysis. The number of markers ranged from 386 SNPs on chromosome 10 to 1018 SNPs on chromosome 1, with an average of 1289 SNPs across chromosomes. The average marker density ranged from 38.61 SNPs/Mb in chromosome 4 to 60 SNPs/Mb in chromosome 10 (Figure 2 and Appendix A). 

The LD decay across the genome was visualized by plotting LD (r^2^) values between adjacent markers against the corresponding physical distance in base pairs (bp) (Figure 3). A notable observation was the rapid decline in LD as the physical distance increased. At a cut-off value of r^2^ = 0.1, the average physical distance was 401 kb.

### 3.3. Marker-Trait Associations for Fe and Zn

The GWAS for Fe and Zn content in rice grains was conducted using two multi-locus models, MLM and FarmCPU. Two SNPs significantly associated with Fe content were detected on chromosomes 1 (S1_34232231) and 11 (S11_2567279). Among the two, S11_2567279 was identified by both models and exhibited the strongest association with Fe content (Figure 4 and Table 1). The SNP S1_34232231 was uniquely identified by FarmCPU.

For the Zn content, the MLM model detected one SNP on chromosome 4 (S4_33308504) with a *p*-value = 2.41 × 10^−4^, while the FarmCPU model detected eight SNPs on chromosomes 3, 4, 5, 7, 11, and 12 (Figure 5 and Table 1). Among the SNPs significantly associated with the Zn content, S4_33308504 on chromosome 4 was detected by both models. 

### 3.4. Candidate Genes for Fe and Zn Content and Their Expression Levels

For the Fe content GWAS, two SNPs on chromosomes 1 and 11 harbored 127 and 120 genes, respectively. For the zinc content, 166, 118, 127, 258, 107, and 121 candidate genes were detected on chromosomes 3, 4, 5, 7, 11, and 12, respectively (Table 2). Among these, the majority fell under the low expressed category, followed by intermediate expressed and then high expressed, based on the in silico analysis of gene expression from RED. Only few genes had unknown expression levels. (Table 2, Appendix A).

We further selected all the candidate genes that are involved in Fe and Zn homeostasis based on functional annotations. A total of 36 genes related to metal homeostasis were identified, with a focus on genes associated with transporter activity and transcription regulatory activity. 

For the Fe content GWAS, two candidate genes tagged by significant SNP S1_34232231 encode for bZIP transcription factor (*LOC_Os01g58760* and *LOC_Os01g59760*), one in the MYB family transcription factor (*LOC_Os01g59660*), and one encodes for transcription factor- TGA5 (*LOC_Os01g59350*) (Table 3). At the locus of SNP S11_2567279, two candidate genes encode for bZIP transcription factor (*LOC_Os11g05640* and *LOC_Os11g06170*) and two are in the transporter family (*LOC_Os11g05390* and *LOC_Os11g05700*). It is interesting to report that two genes (*LOC_Os11g05480* and *LOC_Os05g37170* for Fe and Zn, respectively) encode for transcription factors.

For the zinc content GWAS, 12 candidate genes (*LOC_Os03g25470*, *LOC_Os03g60130*, *LOC_Os03g60850*, *LOC_Os03g61100*, *LOC_Os04g56330*, *LOC_Os04g56470*, *LOC_Os05g37470*, *LOC_Os07g01070*, *LOC_Os07g01560*, *LOC_Os03g60820*, *LOC_Os11g38160*, and *LOC_Os12g05830*) encode for proteins with transporter activity, two encode for proteins with bZIP domains (*LOC_Os07g03220* and *LOC_Os12g06520*) and five encode for MYB (myeloblastosis) transcription factors (*LOC_Os05g37040*, *LOC_Os05g37050*, *LOC_Os05g37060*, *LOC_Os05g37730*, and *LOC_Os07g02800*) (Table 3). The other genes functional annotations included a transcription regulator (*LOC_Os03g25430*), transcription elongation factor (*LOC_Os03g60130* and *LOC_Os12g06850*), transcription termination factor nusG (*LOC_Os03g61030*), AP2-like ethylene-responsive transcription factor AINTEGUMENTA (*LOC_Os04g55970*), AP2-like ethylene-responsive transcription factor PLETHORA 2 (*LOC_Os07g03250*), BEL1-like homeodomain transcription factor (*LOC_Os12g06340*), and E2F family transcription factor protein (*LOC_Os12g06200*) as shown in Table 3.

The in silico expression analysis in rice grain-related tissues showed that among the bZIP family genes, *LOC_Os11g05640* had an intermediate expression level (19.561 FPKM) in the aleurone, *LOC_Os12g06520* had an intermediate expression in the anther and panicle (18.58 and 18.24 FPKM, respectively), and the *LOC_Os11g06170* gene was highly expressed in the pistil (29.41 FPKM) (Table 4). Among the MYB family genes, *LOC_Os01g59660* had the highest expression level of 161.8 FPKM and 37.042 FPKM in the aleurone and anther, respectively; while the *LOC_Os07g02800* gene had an intermediate expression in the panicle, highly expressed in the pistil, and an intermediate expression in the seed, with levels of 15.9, 30.67, and 18.77 FPKM, respectively. The expression levels for all the identified genes are shown in Table 4 and Appendix A.

## 4. Discussion

Genome-wide association mapping is an effective approach that aids in the discovery of genes responsible for controlling specific traits of interest. The efficacy of this method depends on the genetic diversity of the association panels used [42]. In our study, we observed a significant level of phenotypic variation, which was much greater in Fe than in the Zn content in the GWAS panel, as indicated by the descriptive statistics. 

In the present study, we analyzed a set of 85 rice germplasm accessions maintained at the rice breeding program of the INERA, for two important mineral micronutrients, namely, Fe and Zn, in milled rice. The observed range of variability for Fe in this study (0.95–8.68 mg/100 g dwb) was higher than in the previous studies, 0.118 to 0.787 mg/100 g [43] and 0.65 to 2.31 mg/100 g [12]. On the other hand, the range for Zn content (0.87–3.8 mg/100 g dwb) in the present study closely aligns with the range previously reported by Rakotondramanana et al. [44] (1.42 to 4.84 mg/100 g) in whole rice seed. Similar results were also obtained by Descalsota, et al. [45] (0.995 to 2.635 mg/100 g) and Bollinedi et al. [12] (1.3 to 4.62 mg/100 g), in brown rice. This difference could be attributed to the use of different rice accessions in the study [46]. However, compared to other cereals like maize (8.19–25.65 µg/g for Fe and 17.11–43.69 µg/g for Zn) [47] and *Aegilops Tauschii* (30.33–69.44 ppm for Fe and 17.54–49.78 ppm for Zn) [27], rice exhibited high levels of Fe and Zn content in the grain hence the need for biofortification.

The GWAS panel demonstrated a significant variability for the traits studied. Based on our previous work [30], the observed variation among the evaluated rice accessions and the diversity of the panel used highlight the importance of these genetic resources. This significance extends to their potential contributions in crop improvement through breeding and the identification of genes that govern these traits [46].

A total of 8379 high-quality SNPs were obtained from the 85 rice accessions in this diversity panel. This number was relatively higher than what was reported in previous studies by Mogga et al. [48] and Islam et al. [24] which reported 525 high-quality SNPs in 59 rice genotypes and 6565 SNPs in 174 rice accessions. This indicates that the 85 rice accessions used in this study exhibited a high coverage hence being suitable for conducting the GWAS [49].

Association mapping is a population-based study conducted to identify the relation-ships between traits and markers using linkage disequilibrium (LD). In this study, the genome-wide LD decays to r^2^ < 1 within a distance of 401 kilobases (kb). Although the LD decay was relatively slower, these LD decay estimates were lower than the findings of Mather et al. [50], showing long-range LD in temperate japonica rice (>500 kb). However, the LD of our diversity panel had a longer range than the previously published values of 150 kb [51], 109.37 kb, and 214.69 kb in indica rice [52]. The slow LD decay rates in rice are due to its self-pollinating nature, and a relatively small effective population size.

GWAS is designed to evaluate the associations between genotypes and phenotypes [42], and, thus, detect the location of candidate genes. This study utilized two mixed models, to perform GWAS, namely, MLM [53] in which the population structure and kinship were incorporated, and FarmCPU [54] to address false positives by enhancing statistical power and efficiency. In their study, Kaler et al. [55] mentioned that both models are frequently employed in association mapping. A recent study by Bollinedi et al. [12] reported that the MLM has gained widespread popularity for GWAS in crop research, specifically in the context of rice. Various investigations have been conducted on rice to map the QTLs related to Fe and Zn concentration. The QTLs responsible for grain Fe and Zn concentration were identified on different rice chromosomes. It was observed that different rice chromosome positions harbored QTLs associated with one of the micronutrients, revealing their genetic interconnectedness. Swamy et al. [56] reported QTLs on chromosome 1 and 11 for Fe biofortification in rice. A similar finding was observed in our study whereby the significant SNP markers responsible for grain Fe were identified on chromosome 11 using the MLM and FarmCPU models.

Among the SNPs identified, the ones located on chromosome 11 were found to have the highest impact on Fe content, while those on chromosome 4 had the greatest influence on the Zn content in rice. These SNP markers explained a significant portion of the phenotypic variation observed in the respective traits.

This study identified genes belonging to the ZIP and MYB transcription factor families. According to Meng et al. [57], the ZIP family genes play a significant role in Fe and Zn homeostasis in the rice grain. The ZIP family, which includes the zinc-regulated, iron-regulated transporter-like proteins (ZIP) family and iron-regulated transporters (IRTs), plays a crucial role in regulating the absorption and translocation of Zn and Fe in rice. These ZIP transporters are also instrumental in facilitating the cellular uptake and intracellular trafficking of Fe and Zn in plants. Furthermore, they enhance the nutritional content and quality of crops [58]. Moreover, the MYB family genes identified in the current study were reported by Yan et al. [59] who clarified that they are important transcriptional regulators playing key roles in the regulation of plant secondary metabolism, as well as contributing to the regulatory network of anthocyanin biosynthesis. Anthocyanins, which are produced through this biosynthetic pathway, serve to improve plant function under mineral imbalance and also act as metal chelating agents [60]. These gene families are of significant importance in the biofortification of grains since they effectively control Fe and Zn content and the nutritional quality of crops.

## 5. Conclusions

In this study, we conducted a GWAS of Fe and Zn content in rice grains of 85 rice accessions and 8379 DArTseq-derived SNPs. Ten SNPs were significantly associated with Fe and Zn content. Among them, SNP S11_2567279 on chromosome 11 and SNP S4_33308504 on chromosome four were identified for Fe and Zn traits, respectively, using both the MLM and FarmCPU models. Genes belonging to bZIP family genes, MYB family genes, and genes involved in transporter activities were identified within the LD decay window. The identified markers and candidate genes represent valuable resources that can be utilized in rice breeding programs and implemented in Marker Assisted Selection (MAS). Furthermore, these resources can be used to develop improved rice lines with enhanced grain Fe and Zn content, thereby enriching their nutritional value and contributing to enhanced global health outcomes. Additional functional validation of the identified candidate genes will provide deeper insights into their roles in Fe and Zn uptake, transport, and accumulation in rice grains.

## Figures and Tables

**Figure 1 genes-14-01815-f001:**
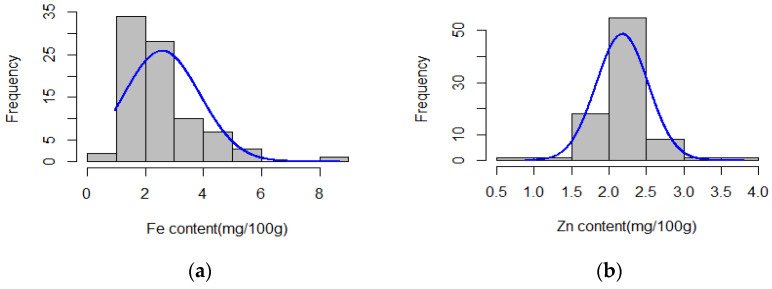
Distribution of accessions based on micronutrient content in grains of rice accessions: (**a**) Fe content; (**b**) Zn content. The blue curves represent the fitted Fe and Zn content data distribution, respectively.

**Figure 2 genes-14-01815-f002:**
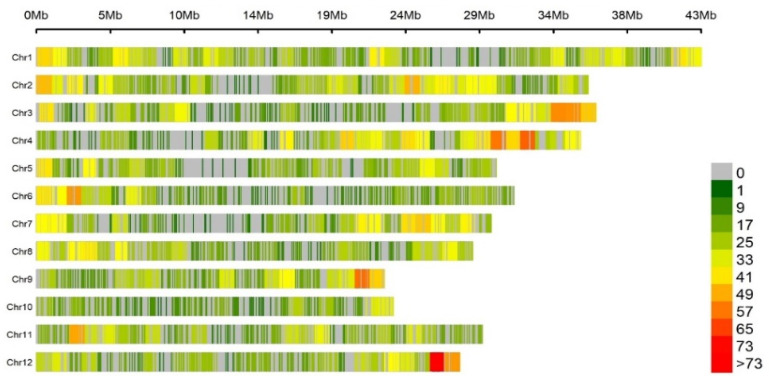
Distribution and density of the 8379 SNP markers along the 12 rice chromosomes. The horizontal axis shows the chromosome length and the SNP density is represented by different colors.

**Figure 3 genes-14-01815-f003:**
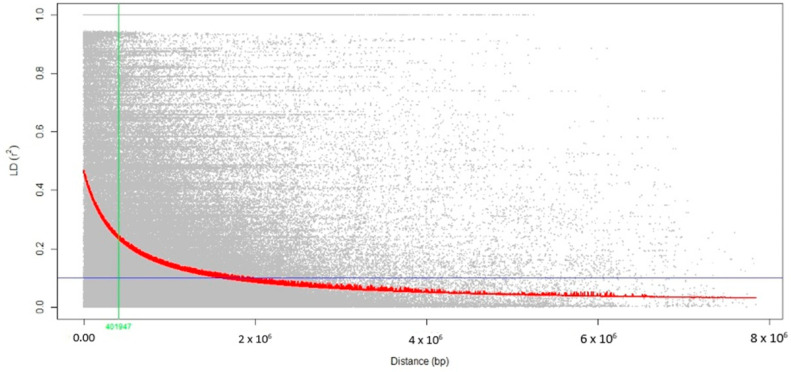
The scatter plot of genome-wide linkage disequilibrium (LD) decay determined based on the r^2^ values of the marker pairs. The red curve line is the regression model fitted to LD decay. The horizontal blue line is the LD at r^2^ = 0.1, whereas the vertical green line is the genome-wide LD decay rate (~401 kb) at r^2^ = 0.1.

**Figure 4 genes-14-01815-f004:**
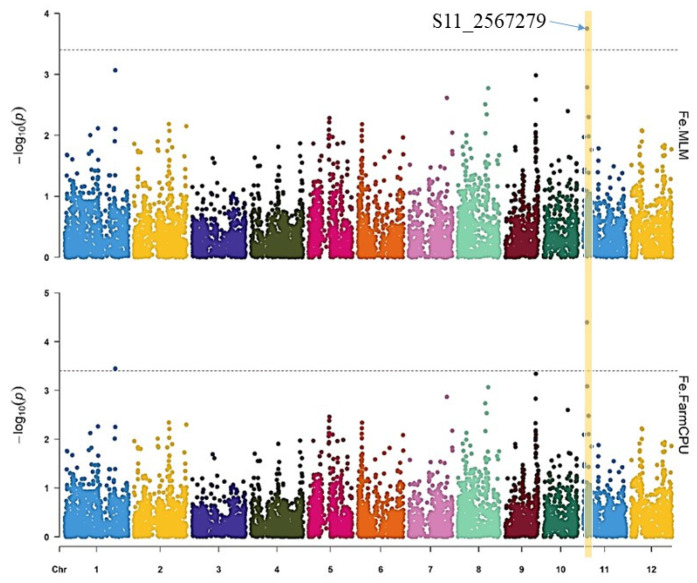
Manhattan plots for the GWAS of Fe content in rice grains using MLM and FarmCPU. The significance threshold (*p* ≤ 4.649 × 10^−4^) is represented by the dashed horizontal line. The x-axis displays the SNP location along the 12 rice chromosomes.

**Figure 5 genes-14-01815-f005:**
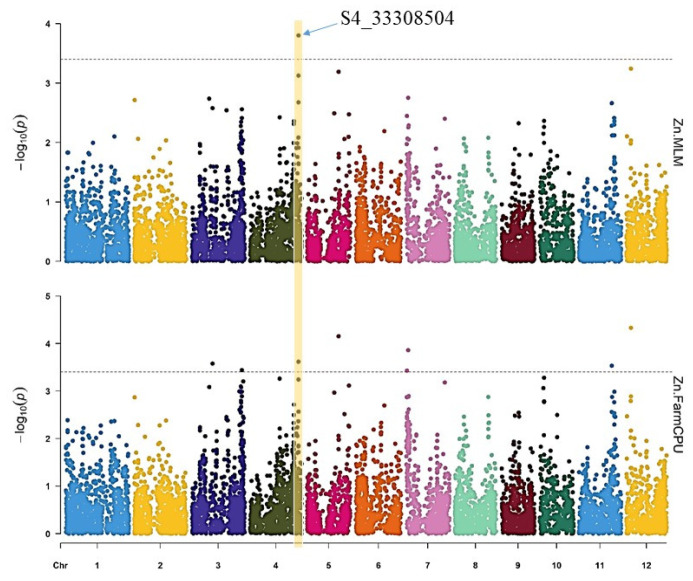
Manhattan plots for the GWAS of Zn content in rice grains using MLM and FarmCPU. The significance threshold (*p* ≤ 4.649 × 10^−4^) is represented by the dashed horizontal line. The x-axis displays the SNP location along the 12 rice chromosomes.

**Table 1 genes-14-01815-t001:** SNP markers significantly associated with Fe and Zn content of rice grains identified in the GWAS.

Trait	Model	SNP	Chr	Position (bp)	*p*-Value
Fe	FarmCPU	S1_34232231	1	34,232,231	3.56 × 10^−4^
Fe	FarmCPU	S11_2567279	11	2,567,279	4.01 × 10^−5^
Fe	MLM	S11_2567279	11	2,567,279	4.01 × 10^−5^
Zn	FarmCPU	S3_14236041	3	14,236,041	2.63 × 10^−4^
Zn	FarmCPU	S3_34414350	3	34,414,350	3.62 × 10^−4^
Zn	FarmCPU	S4_33308504	4	33,308,504	2.41 × 10^−4^
Zn	FarmCPU	S5_21747243	5	21,747,243	7.01 × 10^−5^
Zn	FarmCPU	S7_308113	7	308,113	3.72 × 10^−4^
Zn	FarmCPU	S7_1159472	7	1,159,472	1.37 × 10^−4^
Zn	FarmCPU	S11_22639501	11	22,639,501	2.92 × 10^−4^
Zn	FarmCPU	S12_3069954	12	3,069,954	4.66 × 10^−5^
Zn	MLM	S4_33308504	4	33,308,504	2.41 × 10^−4^

**Table 2 genes-14-01815-t002:** Fe and Zn candidate genes per significant SNP and their corresponding expression levels.

Trait	SNP	Chromosome Position	Number of Candidate Gene Identified				
			Total	Low Expressed	Intermediate Expressed	High Expressed	Unknown
Fe	S1_34232231	1	127	62	33	29	3
	S11_2567279	11	120	57	37	22	4
Zn	S3_14236041	3	22	11	7	3	1
	S3_34414350	3	144	54	40	47	3
	S4_33308504	4	118	57	24	32	5
	S5_21747243	5	127	65	36	20	6
	S7_308113	7	117	56	33	23	5
	S7_1159472	7	141	87	33	19	2
	S11_22639501	11	107	72	18	17	0
	S12_3069954	12	121	66	34	18	3

**Table 3 genes-14-01815-t003:** Candidate genes associated with Fe and Zn homeostasis.

Trait	SNP	Chr	Candidate Gene			Distance (kb)	Annotation
			Gene ID	Start_Pos	End_Pos		
Fe	S1_34232231	1	*LOC_Os01g58760*	33,962,625	33,963,552	268.679	bZIP transcription factor domain containing protein
		1	*LOC_Os01g59350*	34,306,514	34,313,319	−74.283	transcription factor, TGA5, putative, expressed
		1	*LOC_Os01g59660*	34,508,454	34,512,781	−276.223	MYB family transcription factor
		1	*LOC_Os01g59760*	34,565,292	34,568,290	−333.061	bZIP transcription factor, putative, expressed
	S11_2567279	11	*LOC_Os11g05390*	2,413,596	2,417,533	149.746	transporter, major facilitator family
		11	*LOC_Os11g05480*	2,462,254	2,468,733	98.546	transcription factor
		11	*LOC_Os11g05640*	2,560,816	2,563,634	3.645	bZIP transcription factor domain containing protein
		11	*LOC_Os11g05700*	2,605,745	2,610,544	−38.466	ABC transporter family protein, putative
		11	*LOC_Os11g06170*	2,939,742	2,942,834	−372.463	bZIP transcriptional activator RSG
Zn	S3_14236041	3	*LOC_Os03g25430*	14,538,130	14,539,609	−302.089	transcription regulator, putative, expressed
		3	*LOC_Os03g25470*	14,552,198	14,552,749	−316.157	ctr copper transporter family protein
	S3_34414350	3	*LOC_Os03g60130*	34,194,882	34197992	216.358	transcription elongation factor protein
		3	*LOC_Os03g60820*	34,554,642	34,560,824	−140.292	transporter, major facilitator superfamily domain containing protein
		3	*LOC_Os03g60850*	34,574,307	34,576,973	−159.957	peptide transporter PTR2, putative, expressed
		3	*LOC_Os03g61030*	34,671,286	34,674,494	−256.936	transcription termination factor nusG family protein
		3	*LOC_Os03g61100*	34,708,403	34,711,045	−294.053	GDP-mannose transporter, putative
	S4_33308504	4	*LOC_Os04g55970*	33,341,978	33,346,562	−33.474	AP2-like ethylene-responsive transcription factor AINTEGUMENTA, putative, expressed
		4	*LOC_Os04g56330*	33,580,318	33,582,347	−271.814	ABC transporter, ATP-binding protein
		4	*LOC_Os04g56470*	33,661,879	33,664,246	−353.375	amino acid transporter
	S5_21747243	5	*LOC_Os05g37040*	21,646,920	21,647,702	99.541	MYB family transcription factor
		5	*LOC_Os05g37050*	21,650,252	21,651,057	96.186	MYB family transcription factor
		5	*LOC_Os05g37060*	21,654,182	21,655,380	91.863	MYB family transcription factor
		5	*LOC_Os05g37170*	21,720,654	21,724,490	22.753	transcription factor
		5	*LOC_Os05g37470*	21,926,799	21,931,594	−179.556	transmembrane amino acid transporter protein
		5	*LOC_Os05g37730*	22,081,343	22,083,544	−334.1	MYB family transcription factor
	S7_308113	7	*LOC_Os07g01070*	42,657	44,577	263.536	peptide transporter
		7	*LOC_Os07g01560*	348,475	350,594	−40.362	transporter family protein
	S7_1159472	7	*LOC_Os07g02800*	1,046,017	1,048,052	111.42	MYB family transcription factor
		7	*LOC_Os07g03220*	1,267,975	1,268,550	−108.503	bZIP transcription factor domain containing
		7	*LOC_Os07g03250*	1,299,598	1,304,299	−140.126	AP2-like ethylene-responsive transcription factor PLETHORA 2
	S11_22639501	11	*LOC_Os11g38160*	22,625,530	22,627,579	11.922	transporter family protein
	S12_3069954	12	*LOC_Os12g05830*	2,684,760	2,688,178	381.776	transporter-related
		12	*LOC_Os12g06200*	2,939,684	2,945,048	124.906	E2F family transcription factor protein
		12	*LOC_Os12g06340*	3,029,596	3,034,778	35.176	BEL1-like homeodomain transcription factor
		12	*LOC_Os12g06520*	3,153,015	3,156,795	−83.061	bZIP transcription factor domain containing protein
		12	*LOC_Os12g06850*	3,326,815	3,329,114	−256.861	transcription elongation factor protein

**Table 4 genes-14-01815-t004:** The gene expression profiles for Zn and Fe content homeostasis genes.

Gene ID	Annotation	Expression (FPKM)				
		Aleurone	Anther	Panicle	Pistil	Seed
*LOC_Os01g58760*	bZIP transcription factor domain containing protein	0.06	0.94	0.07	2.61	0.62
*LOC_Os01g59350*	transcription factor, TGA5	24.05	11.10	6.22	3.62	0.78
*LOC_Os01g59660*	MYB family transcription factor	161.80	37.04	2.68	4.80	0.94
*LOC_Os01g59760*	bZIP transcription factor	0.21	6.76	5.20	4.46	0.59
*LOC_Os11g05390*	transporter, major facilitator family	0.29	0.32	0.02	0.30	0.99
*LOC_Os11g05480*	transcription factor	0.99	67.84	0.63	1.32	0.85
*LOC_Os11g05640*	bZIP transcription factor domain containing protein	19.56	0.61	0.14	0.11	0.89
*LOC_Os11g05700*	ABC transporter family protein	26.76	2.48	0.81	8.05	0.74
*LOC_Os11g06170*	bZIP transcriptional activator RSG	6.15	11.27	8.74	29.41	0.54
*LOC_Os03g25430*	transcription regulator	3.46	6.64	15.32	0.94	0.78
*LOC_Os03g25470*	ctr copper transporter family protein	1.45	3.55	2.77	0.43	0.60
*LOC_Os03g60130*	transcription elongation factor protein	54.84	28.37	28.78	32.92	0.76
*LOC_Os03g60820*	transporter, major facilitator superfamily domain containing protein	3.23	16.14	29.83	22.14	0.94
*LOC_Os03g60850*	peptide transporter PTR2	32.95	0.38	6.49	4.85	0.84
*LOC_Os03g61030*	transcription termination factor nusG family protein	0.73	2.45	0.64	28.07	0.87
*LOC_Os03g61100*	GDP-mannose transporter	0.13	0.57	0.03	0.01	0.89
*LOC_Os04g55970*	AP2-like ethylene-responsive transcription factor AINTEGUMENTA	52.83	4.09	0.95	0.17	0.93
*LOC_Os04g56330*	ABC transporter, ATP-binding protein	0.00	0.01	0.00	0.00	0.78
*LOC_Os04g56470*	amino acid transporter	18.79	0.42	0.34	1.00	0.96
*LOC_Os05g37040*	MYB family transcription factor	0.00	0.00	0.00	0.00	0.74
*LOC_Os05g37050*	MYB family transcription factor	0.02	0.51	0.29	0.53	0.94
*LOC_Os05g37060*	MYB family transcription factor	0.15	0.10	0.03	0.08	0.91
*LOC_Os05g37170*	transcription factor	4.62	8.86	1.47	5.29	0.61
*LOC_Os05g37470*	transmembrane amino acid transporter protein	0.10	1.10	0.72	3.10	0.77
*LOC_Os05g37730*	MYB family transcription factor	11.14	4.40	14.46	9.07	0.59
*LOC_Os07g01070*	peptide transporter	0.74	0.28	0.26	0.32	0.72
*LOC_Os07g01560*	transporter family protein	68.32	3.04	0.41	16.62	0.81
*LOC_Os07g02800*	MYB family transcription factor	10.58	4.90	15.90	30.67	18.77
*LOC_Os07g03220*	bZIP transcription factor domain containing protein	0.08	0.13	0.00	0.11	0.18
*LOC_Os07g03250*	AP2-like ethylene-responsive transcription factor PLETHORA 2	0.00	0.06	0.05	0.05	0.08
*LOC_Os11g38160*	transporter family protein	6.35	7.59	0.02	0.03	0.81
*LOC_Os12g05830*	transporter-related	12.63	17.54	2.35	9.77	0.54
*LOC_Os12g06200*	E2F family transcription factor protein	1.12	13.01	1.65	4.16	0.75
*LOC_Os12g06340*	BEL1-like homeodomain transcription factor	0.02	0.76	8.60	1.44	0.67
*LOC_Os12g06520*	bZIP transcription factor domain containing protein	9.15	18.58	18.24	14.45	0.28
*LOC_Os12g06850*	transcription elongation factor protein	14.57	12.17	11.82	29.74	0.56
*LOC_Os01g58760*	bZIP transcription factor domain containing protein	0.06	0.94	0.07	2.61	0.62

## Data Availability

The data presented in this study are available on request from the corresponding author.

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
