# Peer review of "Association Mapping of Candidate Genes Associated with Iron and Zinc Content in Rice (Oryza sativa L.) Grains"

_genes, 2023, doi:10.3390/genes14091815_

Round 1

Reviewer 1 Report

Manuscript ID: genes-2575930 Manuscript

Title: Association mapping of candidate genes associated with Iron and Zinc Content in Rice (Oryza sativa L.) Grains This study aimed to identify genomic regions associated with Fe and Zn content in rice grains using SNP markers generated from a set of 85 diverse rice accessions including local landraces. The results of this study present significant assets that can be utilized in breeding and improvement programs aimed at developing rice lines with high grain Fe and Zn content, by integrating the identified markers into a Marker Assisted Selection (MAS) approach. -Comments and Suggestions for Authors Abstract The abstract is well organized. -Please find more corrections as track changes in the manuscript pdf file. Introduction -The introduction section is comprehensive and well written. -Please find more corrections as track changes in the manuscript pdf file. Materials and methods -The Materials and methods section is well written. -Please find more corrections as track changes in the manuscript pdf file. Results & Discussion -The results section is well written. - Please find more corrections as track changes in the manuscript pdf file. Conclusions In conclusions section, the authors must be sure to: -remind readers of your main points. -remind readers of your evidence or arguments. -Wrap everything up by tying it all together. References Please unify the style according to the journal instructions (Please find the corrections in the annotated pdf)

Minor editing of English language required

Author Response

Dear reviewer,

Thank you for your valuable comments and suggestions. We greatly appreciate your thoughtful feedback. Below, we provide responses to each of your comments and the corresponding changes we've made:

1. Comment: homeostasis, Remove the comma

Response: The comma has been removed;

2. Comment: The genetic control for metal transport has been studied 

Response: The verb "have" has been replaced with "has";

3. Comment: replace "The aim of this study was" with "This study aimed to''

Response: The statement "The aim of this study was" has been replaced with "This study aimed to";

4. and contribute to

Response: The word "contributing" has been replaced with "contribute";

5. comment: concentrations

Response: The inclusion of "s"  concentration or to content makes the statement not reader-friendly, Review of papers in the same field does not have "s";

6. Replace In order to with to

Response: The word "In order to" has been replaced with "To";

7. comment: Please revise

Response: The error mentioned has been sorted out;

8. comment: was reported

Response: The verb "was" has been added;

9. comment: remove which 

Response: The word "which" has been removed;

10. comment: long-range

Response: The dash (-) has been added;

11. comment: please correct to "The slow LD decay rates in rice are due to its self-pollinating nature and relatively small effective population size".
-please add a comma after "nature" to separate the two independent clauses. This would make the sentence more readable and easier to understand;

Response: The correction has been done, and a comma has been added after "nature", and now the sentence is more readable and easier to understand;

Thank you once again for your time and careful consideration of our manuscript. We believe that these modifications enhance the clarity and quality of the manuscript.

Best regards,

Chance Bahati Bukomarhe

Reviewer 2 Report

Interesting approach.

Howvere, authirs need to add in introduction and discussion not only local rice varaints, but extend to the other rice acessions from world-wide.

Some minor points:

Line26: contents can be improvement, please, re-formulate.

Lines 28 – 30: not logic. In which cultivar/variant you find these concentrations? Dry or fresh weight?

I would suggest to move lines 49- 57 before lines 43-48. It will be more logic, indeed!

Line 81: must be: „has been studied“

Generaly, in introduction authors need to add  information not only from DRC, but also worldwide.   

Line 125: quantitatively transferred??

Lines 164 – 170: it is not a phenotypic.

Line 202: not a phenotype.

Line 204: „Error! Reference source not found.)”

I am not sure about average for Fe since it is not a Gaussian distribtion

Line 212: please, provide some logical link with previous part (3.1).

Lines 366 – 376: please, made this part more sharp.

Moderate edition

Author Response

Dear reviewer,

Thank you very much for your valuable comments and suggestions. We greatly appreciate your feedback. Below, we provide responses to each of your comments and the corresponding changes we've made:

1. Comment: Line 26: contents can be improvement, please, re-formulate.

Response: This line has been re-formulated

2. Comment: Lines 28 – 30: not logic. In which cultivar/variant you find these concentrations? Dry or fresh weight?

Response: For Iron content, the lowest concentration was found in “Musesekara” and the highest in “08FAN10”. For Zinc, the lowest concentration was found in “ARICA12” and the highest in “Komboka”. Moisture correction had been performed. Reported data is on a dry weight basis, this has now been indicated.

3. Comment: I would suggest to move lines 49- 57 before lines 43-48. It will be more logic, indeed! Generaly, in introduction authors need to add  information not only from DRC, but also worldwide.

Response: The reordering has been done, and we have also added global information to the introduction.

4. comment: Line 125: quantitatively transferred??

Response: Yes, “quantitatively transferred” is used whenever a solution or suspension of precipitate must be transferred from one vessel to another without losing any analyte in the process.

5. comment: Lines 164 – 170: it is not a phenotypic.

Response: Thanks for highliting this, The word phenotypic has been replaced by Fe and Zn content.

6. comment: Line 202: not a phenotype.

Response: The word phenotypic has been replaced by Fe and Zn content

7. comment: Line 204: „Error! Reference source not found.)”

Response: Thank you for highlighting that. The error has been sorted out.

8. comment: Line 212: please, provide some logical link with previous part (3.1).

Response: Part 3.3 provides the logical link between previous parts 3.1 & 3.2.

9. comment: Lines 366 – 376: please, made this part more sharp.

Response: Information on the genes identified has been added, and the part is now sharp. 

Thank you once again for your time and careful consideration of our manuscript. We believe that these modifications enhance the clarity and quality of the manuscript.

Best regards,

Chance Bahati Bukomarhe